# Socio-Demographic Factors Associated with Antibiotics and Antibiotic Resistance Knowledge and Practices in Vietnam: A Cross-Sectional Survey

**DOI:** 10.3390/antibiotics11040471

**Published:** 2022-03-31

**Authors:** Khanh Nguyen Di, Sun Tee Tay, Sasheela Sri La Sri Ponnampalavanar, Duy Toan Pham, Li Ping Wong

**Affiliations:** 1Department of Academic Affairs–Testing, Dong Nai Technology University, Nguyen Khuyen Street, Trang Dai Ward, Bien Hoa City 810000, Vietnam; 2Department of Social and Preventive Medicine, Faculty of Medicine, Universiti Malaya, Kuala Lumpur 50603, Malaysia; 3Department of Medical Microbiology, Faculty of Medicine, Universiti Malaya, Kuala Lumpur 50603, Malaysia; tayst@um.edu.my; 4Department of Medicine, Faculty of Medicine, Universiti Malaya, Kuala Lumpur 50603, Malaysia; sheela@ummc.edu.my; 5Department of Chemistry, College of Natural Sciences, Can Tho University, Can Tho 900000, Vietnam; pdtoan@ctu.edu.vn

**Keywords:** knowledge, practice, antibiotics misuse, antibiotic resistance

## Abstract

(1) Background: Antibiotic resistance (ABR) has been escalating to seriously high levels worldwide, accelerated by the misuse and overuse of antibiotics, especially in Vietnam. In this work, we investigated the Vietnamese public socio-demographic and knowledge factors associated with inappropriate practices of antibiotics to better understand the country’s antibiotic use and ABR. (2) Methods: To this end, a cross-sectional survey was conducted among Vietnamese people aged 18–60 years via Computer Assisted Telephone Interviews (CATIs) from May 2019 to November 2019. (3) Results: Among 3069 responses distributed equally in all 63 provinces in Vietnam, 1306 respondents completed the survey (response rate of 42.5%). Socio-demographically, most participants were male (56.4%), 18–25 years old (40.4%), located in Southern Vietnam (67.1%), highly educated (93.7%), and possessed medical insurance (95.3%). Respondents with higher education levels (college and above) had 2.663 times higher knowledge scores than those with lower education levels (*p* < 0.001). High-income respondents possessed more knowledge than low-income respondents (OR = 1.555, CI 95% 0.835–2.910, *p* = 0.024). Students, non-skilled workers, skilled workers, and professionals and managers had 0.052, 0.150, 0.732, and 0.393 times lower practice scores than the unemployed group, respectively (*p* < 0.001). Furthermore, respondents with higher/adequate knowledge scores had higher practice scores than those with inadequate knowledge scores (*p* < 0.05). (4) Conclusions: The findings indicate that socio-demographic differences in knowledge and practices exist, and focusing on these issues should be the priority in forthcoming interventions. The research data also provide information for policy makers to raise the community’s awareness of ABR.

## 1. Introduction

The overuse of antibiotics is a major contributing factor to the development of antibiotic resistance (ABR), which has been recognized as a global human health threat [1,2,3]. Since new antibiotic development processes take an extensive time for the drugs to be available in the market, numerous infections are becoming difficult to treat [4,5,6]. Up to 10 million deaths from drug-resistant diseases are predicted for 2050 if there is no proper enforcement against ABR [7]. ABR threatens most clinical/public health practices and the economy of both high-income countries and under-resourced countries [2]. Therefore, it is time to take much stronger action to avert this ever-increasing health and economic burden [8].

As end users, the public plays an essential role in antibiotic use and ABR development [9]. An important measurement to minimize the development and spread of resistance is through the rational use of antibiotics [10]. The improper use of antibiotics and ABR may arise from a complex interaction between numerous factors, such as patients’ knowledge and their experience with antibiotics [11,12,13]. Thus, the control of antibiotic utilization needs multifaceted interventions involving knowledgeable and engaged healthcare practitioners and the public [10,12,14,15]. It is therefore important to determine to what extent the community understands antibiotics and how they are used.

In the context of Vietnam, a country with a high rate of ABR, this aforementioned issue is extremely urgent. Vietnam has been facing a high level of ABR as antimicrobials account for over 50% of the total utilized drugs, and are the most popularly dispensed drugs in community pharmacies [16]. In Vietnam, antibiotics are freely available in the pharmacy, and patients can buy them easily. Thus, approximately 88–97% of pharmacies sell antibiotics without prescriptions [16], and 87% of the general public purchases antibiotics in private pharmacies without a doctor’s prescription [17]. Consequently, a high prevalence of bacterial infections and excessive levels of ABR have been observed in Vietnam [9]. For instance, Vietnam had a high level (80.7%) of erythromycin-resistant *Streptococcus pneumoniae* [18], an alarming carbapenem resistance rate (22% and 9%) in *Klebsiella pneumoniae* and *Escherichia coli* isolates [19,20], and a high prevalence (29.5%) of hospital-acquired infections [21]. In addition to the high burden of infectious diseases and relatively unrestricted access to medication, less effective healthcare legislation and inadequate preventive knowledge have posed difficulties to the control and monitoring of antibiotic use, leading to the development of ABR. However, only a few studies have included data on investigating public knowledge of antibiotics and practices in developing countries [5,10,14,21,22]. To our best knowledge, little research has been conducted on this subject in the Vietnamese population.

A better understanding of public knowledge and practice of antibiotic use, as well as their associations with the public socio-demographic characteristics, may assist in tackling ABR in Vietnam. Herein, we reported an investigation on this issue in the Vietnamese public aged 18–60 years regarding antibiotic use and ABR. The results could help establish priorities for antibiotic stewardship policies and reassess their strengths and weaknesses in the implementation of guidelines related to antibiotic use in Vietnam.

## 2. Results

### 2.1. Participant Characteristics

In total, 10,670 telephone numbers were dialed in 63 provinces and cities in Vietnam, of which 3069 were contactable. Among them, 1306 households provided a complete response to the survey, which resulted in a response rate of 42.5%. The reasons received from respondents for not completing the interviews (1763 respondents) were no time/busy, not interested in the topic, refusing without reasons, and ignoring the issues.

Table 1 demonstrates the socio-demographic characteristics of the participants. Among 1306 respondents, a majority of them were female (56.4%), living in the rural area (46.9%) of Southern Vietnam (67.1%), in the age group of 18–25 (40.4%), possessed a high education level of college and above (93.7%), and had a high income of >5 million VND/month (66.1%). Moreover, most of them were skilled workers (28.9%) and had medical insurance (95.3%).

### 2.2. Antibiotic and ABR-Related Knowledge

Figure 1A represents the percentage of participant responses (true/false/don’t know) to the knowledge of general antibiotic information (questions B1–B6), antibiotic usage (question B7–B11), ABR and its consequence (questions B12–B27), and antibiotic side effects (questions B28–B31). For all 31 questions, the respondents who answered correctly had the highest rate (>65%). Significant associations (multivariate analysis, *p* < 0.05) were found between respondents’ knowledge and their respective gender, age, education level, occupation, and monthly income (Table 2). To this end, females were more likely to have better knowledge (OR 1.204, CI 95% 0.933–1.554, *p* = 0.027) than males. The young age group of 18–25 years (OR 1.315, CI 95% 0.789–1.864), 26–35 years (OR 2.072, CI 95% 1.103–3.513), and 36–45 years (OR 1.370, CI 95% 0.890–1.954) had higher chances of possessing high antibiotic knowledge scores compared to the oldest age group of 46–<60 years. Respondents with higher education levels (college and above) had 2.663 times higher knowledge scores than those with lower levels (*p* < 0.001). High-income respondents possessed more knowledge than low-income respondents (OR = 1.555, CI 95% 0.835–2.910, *p* = 0.024). Students had higher chances of better knowledge of antibiotics (OR 1.774, CI 95% 1.010–2.238) compared to unemployed people. Interestingly, non-skilled workers, skilled workers, professionals, and housewives possessed less antibiotic knowledge than the unemployed respondents.

### 2.3. Practices Regarding Antibiotic Use

Regarding the participant practice levels for antibiotic usage and ABR, a majority of participants correctly answered the questions/statements in all questions from D1 to D13 (Figure 1B). An exception was noted for question D5, which only 47.7% of respondents answered correctly. The total possible practice score was 39, and a high score of ≥20 was considered good practice. The practice score was significantly correlated with the respondent’s occupation, as students, non-skilled workers, skilled workers, and professionals and managers had 0.052, 0.150, 0.732, and 0.393 times lower practice scores than the unemployed groups, respectively (*p* < 0.001) (Table 2). On the other hand, housewives were more likely to have better practices on antibiotic use (OR 2.344, CI 95% 1.502–3.617) than unemployed people.

In terms of the relationships between the public knowledge and their practices on antibiotic use, a significant correlation was found, where respondents with lower/inadequate knowledge scores (OR 0.257, CI 95% 0.034–0.898) possessed lower practice scores than those with higher/adequate knowledge scores (*p* < 0.05).

## 3. Discussion

This work is the first study to investigate the public antibiotics and ABR knowledge and practices in Vietnam, and their associated socio-demographic factors, using the CATI approach. CATI was chosen because it would help improve data quality with minimal mistakes, cost reduction, and efficiency, and is less time-consuming in terms of data transferring after the interviews for processing of data analysis [23].

In terms of public knowledge of antibiotic use and ABR, although a majority of participants agreed that antibiotics were used to treat infections caused by bacteria (67.2%), they also stated that antibiotics were used to treat infections caused by viruses (67.9%). This ratio was higher than in a similar study (44.1%) in China and other countries [24,25,26]. This showed that the respondents mistakenly believed that antibiotics work for both viruses and bacteria, or that they cannot distinguish between bacterial and viral infections. Thus, further educational campaigns to enhance public knowledge of antibiotic use and ABR are necessary. 

Regarding practices on antibiotic use, a common misconception among respondents was that when they had a fever, they took antibiotics with the hope of quickly recovering from the disease. This led to the result that many fevers caused by viruses had been treated with antibiotics, thus increasing the development of ABR. Moreover, although most respondents agreed that they would consult a doctor before starting a course of antibiotic treatment (36.2% strongly agreed, 23% agreed), in agreement with a report in South Korea (46.9%) [27], approximately 40% of the respondents did not consult a doctor before purchasing antibiotics. This alarming fact demonstrates the crucial need for interventions, especially in terms of education, from the government. In reality, the respondents who bought antibiotics at pharmacies without a prescription accounted for a very high proportion (39% strongly agreed, 20.7% agreed), much higher than in a study in Namibia (only 15%) [28]. This is due to people not obeying laws in Vietnam that restrict the public from self-medicating with antibiotics at pharmacies. Additionally, in the Vietnamese context, pharmacies selling over-the-counter antibiotics are very common. Interestingly, 86.1% of respondents said that antibiotics should not be bought arbitrarily without a prescription from a doctor, much higher than the rates in Greece (44.6%) [25] and Jordan [26]. This information contradicts the fact that 87% of the general public purchase antibiotics from private pharmacies without a doctor’s prescription [17]. The reason might be due to convenience; since people know that antibiotics should be bought with a prescription, they do not want to waste time consulting doctors and instead freely buy the drugs at a local pharmacy. Thus, policy makers should focus on this issue to decrease the prevalence of antibiotic abuse and ABR. Moreover, the pharmacists’ responsibilities should be emphasized and critically monitored in the near future. Since they are the main distributors of antibiotics to patients, policies on enhancing their roles and awareness in ABR should be employed. 

Interestingly, most respondents (>65%) agreed to both opposite statements to intentionally use a lower or higher antibiotic dose than prescribed by the doctor. This demonstrated that, depending on the disease, respondents arbitrarily adjusted the antibiotic dosage. Strict policies should be enforced to minimize this problem, since incorrect dosing could dramatically lead to the development of ABR. Lastly, although most people (65.4%) agreed that one should check the expiry date and follow the instructions before using antibiotics, this number was significantly different from a study result in Malaysia (92.2%) [29]. Since expired drugs might contain toxicity [30] it is necessary to change people’s habits on this issue.

Analysis of the associations between the knowledge/practices and socio-demographic characteristics revealed that a respondent’s occupation affects all scores, whereas gender, age group, education level, and monthly income affect only the knowledge level. Expectedly, education level is a factor influencing a respondent’s knowledge. This can be explained by different levels of awareness integrated with basic and advanced knowledge they learned in school [31]. This finding is also consistent with the occupational level variable, which was found to have a significant association with a respondent’s knowledge (i.e., students had the highest knowledge). Regarding the socio-demographic characteristics and ABR practice, interestingly, housewives had higher practice scores than the unemployed, whereas other occupations of students, workers, skilled workers, and professionals and managers had a risk of possessing lower practice scores than the unemployed respondents. This can be because women who take care of children and family members tend to receive advice from pharmacy staff.

Interestingly, this study found a good correlation between respondents’ level of knowledge and their corresponding practice level, where respondents with high knowledge scores also had higher practice scores. This finding supports that better knowledge enhances the appropriate utilization of antibiotics [31]. This evidence, together with the association between practices on antibiotic use and the participant occupations, emphasizes that communication interventions that target the general public regarding their occupations are necessary to fill the knowledge gaps. This also indicates the importance of healthcare education and educational campaigns to improve public knowledge, consequently enhancing practices on antibiotic use and awareness of ABR. 

### Limitations of the Study

Although new information regarding Vietnamese public antibiotic usage and ABR has been derived from this study, some limitations were noted. Firstly, the study findings may not represent the whole population of Vietnam since the respondent rate was mostly the younger generation in the age groups of 18–25 and 26–35 (75.9%) years old, with high education (93.7%). Thus, it would be an overestimation of the respondents’ knowledge of the issue. Secondly, the CATI method might yield selection bias, and some people might not want to disappoint the surveyor, thus influencing the accuracy of questions related to, e.g., sharing antibiotics or not adhering to prescription instructions. Thirdly, the survey tool, although critically written, revised, and validated, did not have adequate neutral response, and some of the questions are leading. It is suggested that further research on a similar topic in the future could combine different approaches to maximize the response rates and accuracy. Likewise, longitudinal studies should be further employed to gain reliable data on antibiotic use and ABR in Vietnam.

## 4. Materials and Methods

### 4.1. Participant Recruitment

A cross-sectional study targeting Vietnamese people aged 18–60 years who lived in households with landline telephone numbers was conducted using the CATI method [23], based on Vietnam’s national telephone directory, between May 2019 and November 2019 (6 months). According to the nationwide distribution of the population of all 63 provinces and cities in Vietnam, the sample was stratified by provincial territories to ensure geographical illustration and generalization.

Participants included Vietnamese nationals, aged between 18 and 60 years old and a resident of the contacted household, who agreed to provide verbal informed consent to participate in the interview. Only one person per household was surveyed. The telephone numbers were generated randomly using Research Randomizer software. The Cochran formula was used to calculate the sample size, based on the most conservative expected rate of ABR of 50%, with a 3% of margin of error and a 95% confidence level. Hence, for the target group of approximately 54,823,000 (the estimated number of the investigated population), the required sample for this study was 1067. Assumedly, the response rate of total calls was 10%. Hence, 10,670 telephone calls were made, divided proportionally among all 63 cities and provinces in Vietnam.

### 4.2. Instruments

The questionnaire encompassed five sections with a total of 53 questions. The first section consisted of nine questions regarding the socio-demographic characteristics of the respondents. The second and third sections comprised 31 and 13 questions/statements regarding the knowledge (B1–B31) and practice (D1–D13), respectively, towards antibiotic use and ABR among the general public in Vietnam. To avoid selection bias, some questions/statements had positive meaning and others were negative. The full list of the questions/statements is shown in Table 3. The respondents were required to answer all questions.

The answer choices were “True”, “False”, and “Don’t know” for the knowledge, which was scored as 0 (for No/Don’t know answer) and 1 (for Yes answer) for the positive items, and vice versa for the negative items. On the other hand, a four-point Likert scale, 1—“Strongly disagree”/”Never”, 2—“Disagree”/”Rarely”, 3—“Agree”/”Sometimes”, and 4—“Strongly agree”/”Often”, was employed for the practice section. The positive items were scored 3, 2, 1, and 0 for responses of 1, 2, 3, and 4, respectively. The negative questions were reversely scored. A cutoff percentage of 65% was considered to be the threshold for the appropriate knowledge. The “Strongly agree” and “Agree” responses, and “Strongly disagree” and “Disagree” were then grouped for comprehensive analyses.

The questionnaire was initially developed in English and then translated into Vietnamese. The translated version was reviewed by secondary translators and independent language experts to ensure the accuracy of the translation. Then, the back-translated version was double-checked by researchers for further modifications and improvements. The questionnaire was also validated by a panel of experts including physicians, medical microbiologists, academicians, and infectious disease specialists. A pilot test with 20 eligible participants was additionally conducted to assess the interviewees’ understanding of the sentence structure and wording. Then, the received feedback was reviewed and adapted to modify the questionnaire ahead of the actual data collection.

### 4.3. Ethical Approval

All participants were informed of the aims and objectives of the study. Verbal consent was given by the respondents at the beginning of the interviews. The study was ethically approved by the University of Malaya Research Ethics Committee (Non-Medical) (UM.TNC2/UMREC-594).

### 4.4. Statistical Analyses

Tests for normal distribution or normality tests were performed to see whether the continuous data were normally distributed, by visually checking and testing the kurtosis and skewness of normal distribution, using the Statistical Package for the Social Sciences (SPSS; Chicago, IL, USA) [23]. Parametric tests were also used to determine normal distribution. Multivariable logistic regression analysis was utilized to assess the relationship between the demographic factors influencing the level of public knowledge/practice and their socio-demographic characteristics. Variables with statistically significant correlations (*p* < 0.05) in the univariate analyses were multivariate analyzed by the forced-entry method. Odds ratios (OR), confidence intervals (CI 95%), and *p*-values were calculated and reported, with significance at *p* < 0.05.

## 5. Conclusions

This cross-sectional study provides a more comprehensive understanding of the current public knowledge and customary practices toward antibiotic usage and the present ABR status in Vietnam, highlighting important gaps in knowledge of antibiotic malpractices in this country. The findings indicate that participants’ occupation and their respective knowledge levels significantly correlate with their practices on antibiotic use. As socio-demographic differences in knowledge and practices of antibiotic use and ABR exist, focusing on these issues should be the priority in forthcoming interventions. The research findings also contribute to the need to raise the community’s awareness of ABR and provide information for policy makers to guide future improvement, revision, and reformation of policies regarding this urgent issue.

## Figures and Tables

**Figure 1 antibiotics-11-00471-f001:**
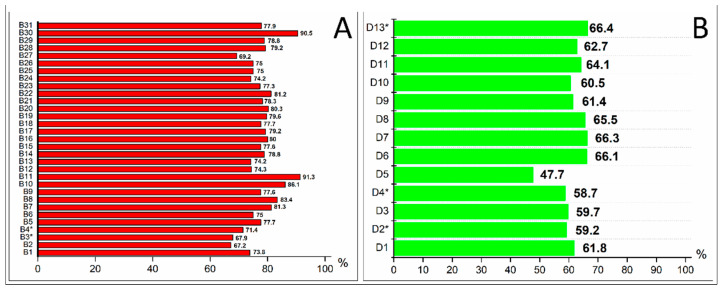
Percentages of correct respondent responses on Vietnamese public knowledge (**A**) and practice (**B**) regarding antibiotic use and resistance (*n* = 1306). The full questions/statements (B1–B31 and D1–D13) are shown in Table 3; *, the correct response to these questions is False/Don’t agree.

**Table 1 antibiotics-11-00471-t001:** Socio-demographic characteristics of the respondents in Vietnam (*n* = 1306).

Socio-Demographic Characteristics	No. Response (*n*)	Percentage (%)
*Gender*		
Female	570	43.6
Male	736	56.4
** *Age (years old)* **		
18–25	528	40.4
26–35	463	35.5
36–45	155	11.9
46–<60	154	11.8
** *Education level* **		
Low (below secondary school)	82	6.3
High (college and above)	1224	93.7
** *Monthly income* **		
<5 million VND	443	33.9
>5 million VND	863	66.1
** *Geographical area* **		
Northern Vietnam	212	16.2
Central Vietnam	218	16.7
Southern Vietnam	876	67.1
** *Living area* **		
Urban	355	27.2
Sub-urban	339	26
Rural	612	46.9
** *Occupation* **		
Student	118	9.0
Non-skilled worker	312	23.9
Skilled worker	377	28.9
Professional and managerial	222	17.0
Housewife	80	6.1
Unemployed	197	15.1
** *Insurance* **		
With medical insurance	1254	95.3
Without medical insurance	61	4.7

**Table 2 antibiotics-11-00471-t002:** Multivariate analysis on the associations between Vietnam public knowledge/practice and socio-demographic characteristics. OR: odds ratio; CI: confidence intervals.

Variables	Knowledge		*p*-Value	OR (CI 95%)	Practice		*p*-Value	OR (CI 95%)
	Adequate (Score ≥ 16)	Inadequate (Score < 16)			Adequate (Score ≥ 20)	Inadequate (Score < 20)		
**Gender**			**0.027**					
Female	26.0%	74.0%		**1.204 (0.933–1.554)**	6.7%	93.3%		0.673 (0.256–1.103)
Male	22.6%	77.4%		**1**	5.8%	94.2%	0.126	1
**Age**								
18–25	22.2%	77.8%		**1.315 (0.789–1.864)**	7.8%	92.2%		0.391 (0.112–0.670)
26–35	19.2%	80.8%		**2.072 (1.103–3.513)**	2.8%	97.2%		0.454 (0.201–1.240)
36–45	30.3%	69.7%		**1.370 (0.890–1.954)**	7.7%	92.3%		0.339 (0.101–0.790)
46–<60	37.7%	62.3%	**0.008**	**1**	9.7%	90.3%	0.903	1
**Education level**								
High	43.9%	56.1%		**2.663 (1.688–4.202)**	12.2%	87.8%		0.937 (0.454–1.322)
Low	22.7%	77.3%	**<0.001**	**1**	5.8%	94.2%	0.889	1
**Occupation**								
Student	38.1%	61.9%		**1.774 (1.010–2.238)**	23.7%	76.3%		**0.052 (0.003–0.230)**
Non-skilled worker	17.6%	82.4%		**0.563 (0.222–1.134)**	11.2%	88.8%		**0.150 (0.067–0.381)**
Skilled worker	24.7%	75.3%		**0.705 (0.311–1.458)**	1.9%	98.1%		**0.732 (0.334–1.222)**
Professional	26.6%	73.4%		**0.654 (0.214–1.201)**	3.6%	96.4%		**0.393 (0.011–1.144)**
Housewife	21.2%	78.8%		**0.852 (0.469–2.080)**	0.0%	100%		**2.344 (1.502–3.617)**
Unemployed	22.8%	77.2%	**0.005**	**1**	1.5%	98.5%	**0.000**	1
**Monthly Income**								
>5 million VND	24.6%	75.4%		**1.555 (0.835–2.910)**	10.4%	89.6%		1.576 (0.760–2.459)
<5 million VND	23.8%	76.2%	**0.024**	**1**	4.1%	95.9%	0.243	1
**Geographical area**								
Northern Vietnam	30.7%	69.3%		0.747 (0.291–1.578)	9.0%	91.0%		0.749 (0.320–1.805)
Middle of Vietnam	25.2%	74.8%		1.006 (0.549–1.600)	7.8%	92.2%		0.670 (0.178–1.460)
Southern Vietnam	22.1%	77.9%	0.261	1	5.1%	94.9%	0.404	1
**Living area**								
Urban	23.1%	76.9%		0.171 (0.049–0.410)	9.0%	91.0%		0.712 (0.281–1.720)
Sub-urban	25.4%	74.6%		0.446 (0.120–1.095)	6.2%	93.8%		0.877 (0.336–1.572)
Rural	23.9%	76.1%	0.379	1	4.6%	95.4%	0.494	1
** *Insurance* **								
With insurance	26.2%	73.8%		1.130 (0.629–2.028)	9.8%	90.2%		0.708 (0.291–1.460)
Without insurance	23.9%	76.1%	0.390	1	6.0%	94.0%	0.452	1
** *Knowledge* **								
Inadequate					8.6%	91.4%		**0.257 (0.034–0.898)**
Adequate					3.2%	96.8%	**0.002**	**1**

**Table 3 antibiotics-11-00471-t003:** List of all knowledge (B1–B31) and practice (D1–D13) questions/statements regarding antibiotic use and ABR among the public in Vietnam.

Item	Question/Statement	Item	Question/Statement
**KNOWLEDGE (B1–B31)**		**B24**	Patients with antibiotic-resistant infections require a longer recovery period
**B1**	Common cold and flu are caused by viruses, not by bacteria	**B25**	Treatment for antibiotic-resistant infection is more expensive
**B2**	Antibiotics are used to cure infections caused by bacteria only	**B26**	More serious illnesses can develop with an antibiotic-resistant infection
**B3**	Antibiotics are used to cure infections caused by viruses	**B27**	More doctor visits are required with an antibiotic-resistant infection
**B4**	Antibiotics speed up the recovery from most coughs and colds	**B28**	Some antibiotics may cause side effects such as diarrhea, vomiting, and headache
**B5**	Different types of antibiotics are used to cure different diseases	**B29**	Some antibiotics may cause allergic reactions such as rash, shortness of breath, and swelling of the lips or tongue
**B6**	The human body can fight against mild infections without antibiotics	**B30**	One should consult a doctor when experiencing the above antibiotic side effects
**B7**	One should never save antibiotics for future use	**B31**	The use of some antibiotics can cause an imbalance in gut microorganisms
**B8**	One should never use leftover antibiotics from previous treatments	**PRACTICE (D1–D13)**	
**B9**	One should never share leftover antibiotics with other people	**D1**	I either take antibiotics or ask the doctor to prescribe antibiotics when I have a common cold, cough, and/or flu-like symptoms
**B10**	One should never buy antibiotics without a doctor’s prescription	**D2**	I consult a doctor before starting a course of antibiotics
**B11**	One should complete the dose of antibiotic prescribed by a doctor	**D3**	I get antibiotics at the pharmacy store without a prescription
**B12**	Infections caused by antibiotic-resistant bacteria are increasing in the community	**D4**	I complete the full course of antibiotics prescribed by a doctor
**B13**	Antibiotic resistance means bacteria are not controlled/killed by antibiotics anymore	**D5**	I discontinue taking antibiotics when symptoms have improved or resolved, even if I have not completed the recommended course of treatment
**B14**	Taking antibiotics unnecessarily or without doctor’s prescription may contribute to the development of antibiotic resistance	**D6**	I intentionally use a lower dose of antibiotics rather than the recommended one by a doctor
**B15**	Taking antibiotics without doctor’s prescription can contribute to the development of antibiotic resistance	**D7**	I intentionally use a higher dose of antibiotics rather than the recommended one by a doctor
**B16**	Infection caused by antibiotic-resistant bacteria cannot be easily cured	**D8**	I fail to comply with the recommendation by a doctor (i.e., missed dose, accidentally overdose)
**B17**	Taking a complete dose of antibiotics can cure the bacterial infection and prevent antibiotic resistance	**D9**	I use leftover antibiotics from my previous treatments without seeking medical advice if I develop similar symptoms
**B18**	Taking an incomplete dose of antibiotics can lead to infection not completely cured or a relapse of the disease	**D10**	I share leftover antibiotics with others
**B19**	Leftover antibiotics are not a complete dose, hence, are not able to eliminate a bacterial infection	**D11**	I am going to another doctor if the present doctor refuses to give me antibiotics for my medical treatment
**B20**	People can act as carriers of antibiotic-resistant bacteria and spread the infection to close contacts (family members or friends)	**D12**	I keep antibiotics at home for an emergency case for my children
**B21**	Animals can act as carriers of antibiotic-resistant bacteria and spread the infection to humans	**D13**	I look at the expiry date, read and follow the instructions label of the antibiotics before taking them
**B22**	Animal products (meat, eggs) can be a source of antibiotic-resistant bacteria		
**B23**	Good personal hygiene can reduce the spread of antibiotic-resistant bacteria in the community		

## Data Availability

Not applicable.

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
