# Peer review of "Socio-Demographic Factors Associated with Antibiotics and Antibiotic Resistance Knowledge and Practices in Vietnam: A Cross-Sectional Survey"

_antibiotics, 2022, doi:10.3390/antibiotics11040471_

Round 1

Reviewer 1 Report

The manuscript “Factors influencing antibiotics overuse and misuse: A cross-sectional survey in Vietnam” focus on a very important issue regarding public health.

However, there are some concerns about the structure and contents of the manuscript.

Line 2: the title doesn’t seem very appropriate to the content of the study. In my perspective, the socio-demographic factors studied do not influence antibiotics overuse or misuse. As the authors say in lines 151 and 171 “…in the Vietnamese context, pharmacies selling over-the-counter antibiotics have been very common” and “This was due to not-obeyed laws in Vietnam to restrict people to self-medicate with antibiotics at pharmacy store.”

In my opinion, “Socio-demographic factors associated to public antibiotics and ABR knowledge and practices in Vietnam: A cross-sectional survey” seems a much more adequate title.

Lines 30-32: “: Findings are essential for public health professionals in the formulation of better plans of action against ABR in Vietnam”. These are not conclusions form the study. This is the usefulness of the study.

Line 80: The materials and methods section should be presented before the results.

Lines 84-87. The authors say that 1763 respondents did not conclude the interviews in part because “do not know about antibiotics and ABR”. This is very important and may influence the results, since part of the questionnaire was specifically about antibiotics knowledge. From the 1763 respondents, the ones who answered “do not know about antibiotics and ABR” should be included in the 2.2 section.

Line 119: What the authors means with “specialist”? This respondent’s occupation is not in tables 2 or 3.

Lines 212-213: As the authors says: “the CATI method might yield selection bias with self-selection of interviewees to answer the questionnaires”. This causes concerns about the credibility of the study and should be corrected.

Author Response

Responses to the reviewers’ comments

Line 2: the title doesn’t seem very appropriate to the content of the study. In my perspective, the socio-demographic factors studied do not influence antibiotics overuse or misuse. As the authors say in lines 151 and 171 “…in the Vietnamese context, pharmacies selling over-the-counter antibiotics have been very common” and “This was due to not-obeyed laws in Vietnam to restrict people to self-medicate with antibiotics at pharmacy store.”

In my opinion, “Socio-demographic factors associated to public antibiotics and ABR knowledge and practices in Vietnam: A cross-sectional survey” seems a much more adequate title.

Thank you for your suggestion. Following the suggestion, the title has been adjusted to “Socio-demographic factors associated to public antibiotics and antibiotics resistance knowledge and practices in Vietnam: A cross-sectional survey”

Lines 30-32: “: Findings are essential for public health professionals in the formulation of better plans of action against ABR in Vietnam”. These are not conclusions form the study. This is the usefulness of the study.

Thanks for your comment. The conclusions have been re-written accordingly.

Line 80: The materials and methods section should be presented before the results.

Thank you. Due to the journal format that requires the Materials and methods section should be after the Discussion section, we must apologize for not following your suggestion on this issue.

Lines 84-87. The authors say that 1763 respondents did not conclude the interviews in part because “do not know about antibiotics and ABR”. This is very important and may influence the results, since part of the questionnaire was specifically about antibiotics knowledge. From the 1763 respondents, the ones who answered “do not know about antibiotics and ABR” should be included in the 2.2 section.

Thanks for your notification. We apologize for confusing you. This is the translation error. The phrase “do not know about antibiotics and ABR” means that the respondents simply refused to complete the questionnaire because they said that they do not give their attention to these issues (i.e., they ignore it). The term “to ignore” and “to not know” is the same in Vietnamese language (“không biết”). Thus, we have revised the phrase to be clearer. Thanks again for mentioning this issue.  

Line 119: What the authors means with “specialist”? This respondent’s occupation is not in tables 2 or 3.

Thanks so much! We made a mistake here. The term “specialist” means skilled workers. Thus, we have adjusted all these words accordingly in the entire manuscript.

Lines 212-213: As the authors says: “the CATI method might yield selection bias with self-selection of interviewees to answer the questionnaires”. This causes concerns about the credibility of the study and should be corrected.

Thank you for your comment. The statement has been adjusted accordingly.

Reviewer 2 Report

Thank you for the opportunity to provide this review.  The authors describe the results of a telephonic survey of Vietnamese people regarding knowledge and practice of antibiotic use/misuse.  The response rate was high enough to draw conclusions and these data are helpful in designing education efforts.  I thank the authors for pursuing a topic such as this. Please find comments below.  

General:

Some places the wording is difficult to follow with run on sentences. Would recommend an overall editing to improve the readability. 

Introduction:

Primary points are covered.

It's a little long, I would reduce the general statements in the first two paragraphs and get to the primary point as most readers understand the dangers of AMR.

Methods (after discussion):

Mostly clear to reader.

I would include clearly what was considered appropriate knowledge.  In the results it becomes clearer that a cutoff of 65% was used. (Line 116/117 in results should be in methods - can be reiterated in results)

Make it clear which responses would be grouped (e.g. strongly agree plus agree) for analyses.

Why did you chose to dichotomize the knowledge as opposed to keep it as a continuous variable score?  I think the actual scores should be displayed and reported overall and by each patient characteristic evaluated as well, not as replacement. 

Could respondents chose not to answer/defer?  Please clarify. 

Some limitations to the survey tool based on actual questions/language are listed in comments below (discussion section).

Results:

You state that there was an even spread over geographic regions of Vietnam, but it appears that the overwhelming majority were in Southern Vietnam.  It doesn't make this incorrect necessarily, but does seem a bit contradictory.

Sometimes the words "most" or "majority" is used to describe things that are less than 50%.   Line 90 is example

Please include confidence intervals for all of your ORs.  

Most of the time 2 decimal points is acceptable for reporting ORs. 

In the text, please indicate if these ORs were multivariate or univariate regression.  (Beginning line 101)

As mentioned above, please provide actual knowledge scores, not just "adequate/inadequate".  

Tables/Figures:

I would move the survey tool to the end or include as a supplemental, it interrupts the flow of the paper currently.

B13 in the survey questions, controlled is misspelled

Not your fault per se, but Tables 2 and 3 would be much better if they were on a single page each for layout for ease of reading.

Please indicate in Table 3 if this is univariate or multivariate; also include confidence intervals

Discussion:

Beginning at line 150, you talk a lot about over-the-counter pharmacy sales.  This is fine, but it seems to be out of place in this initial paragraph.  You quote a % of OTC purchases that was from a different study (156-158), then later  (170) you quote your current data.  I would place these type of statements together to improve flow through this section.  There are some other opportunities to improve flow through these 2 paragraphs.

Also, opportunity to compare self-medication to other studies.  Haiti, PMID: 29138650; Tanzania, PMID: 32599699 (investigators actually went to community pharmacies to evaluate OTC).  I think leveraging pharmacies is an important discussion point to include that could be a next step for your research.  

Is antibiotic quality a concern in Vietnam?  Please state either way.

Line 165-167: You state the majority of patients seek advice before getting an antibiotic.  However that 59% means 41% do not, that's concerning.  I would reverse the way this is stated and focus on the 41%, not the 59%.  Again will help flow of paragraphs here.

What is the role of pharmacies in Vietnam?  I would spend some more time discussing pharmacists/pharmacy role/responsibilities - now and perhaps future.  The questions infer there is limited impact of a pharmacist/pharmacy employee.  This would help the reader understand the landscape a bit better.

Are antibiotics freely available in a pharmacy or have to distributed by a pharmacist (or pharmacy employee)?  That impacts access too

Some of the characteristics that are linked to knowledge likely don't matter that much.  For example, while females scored statistically higher, adequate knowledge was in 26% vs 23% in females versus males.  I would argue that it doesn't really matter what's statistically associated when both are far lacking in knowledge based on your assessment.  Unless there were major differences in median scores. Please update the language to reflect this.  

The survey tool itself has some major limitations that need to be mentioned.  - Many of the knowledge questions are very leading using the word "never" consistently.  This creates potential bias in answers. 

- The survey did not appear to have a neutral response or an opportunity to not answer or defer on the question. That's a minor limitation.

- My experience has been that in these types of surveys people do not want to disappoint the surveyor and thus the accuracy of such questions like sharing antibiotics or not adhering to prescription instructions can be influenced.  However, it doesn't appear to be in your study, but still a potential limitation.

- Does using households with a landline impact distribution at all? 

- Questions like D1 are a bit concerning of the overall message.  While I know the authors discourage self-prescribing, and thus self-diagnosis, this question actual calls for the respondent to self diagnosis.  Unfortunately it reflects a bit of the problem.  I think that question itself is a limitation.

Author Response

To the reviewer #2

General:

Some places the wording is difficult to follow with run on sentences. Would recommend an overall editing to improve the readability.

Introduction:

Primary points are covered.

It's a little long, I would reduce the general statements in the first two paragraphs and get to the primary point as most readers understand the dangers of AMR.

Thanks for your recommendation. The article wordings, grammars, and format have been critically revised.

Methods (after discussion):

Mostly clear to reader.

I would include clearly what was considered appropriate knowledge.  In the results it becomes clearer that a cutoff of 65% was used. (Line 116/117 in results should be in methods - can be reiterated in results)

Make it clear which responses would be grouped (e.g. strongly agree plus agree) for analyses.

Why did you chose to dichotomize the knowledge as opposed to keep it as a continuous variable score?  I think the actual scores should be displayed and reported overall and by each patient characteristic evaluated as well, not as replacement.

Could respondents chose not to answer/defer?  Please clarify.

Some limitations to the survey tool based on actual questions/language are listed in comments below (discussion section).

Thank you for your suggestion. The Introduction has been revised accordingly.

Thanks for your thoughtful comments.

- The cutoff of 65% was added in the Methods section accordingly.

- The grouped responses were denoted clearly.

- Regarding the knowledge score, we apologize in case we misunderstand your point, in our humble opinion, we think that it is better to dichotomize the results. It was due to the fact that the score of each respondent were different, and the number of respondents was high, thus, the continuous score presentation would be redundant and confusing, with lots of numbers. Conclusively, we thought that the grouping of this variable should be appropriate. 

- The respondents must answer all questions accordingly. This was mentioned and explained carefully to all participants prior to the CATI interview. The information has been added in the Methods section.

Results:

You state that there was an even spread over geographic regions of Vietnam, but it appears that the overwhelming majority were in Southern Vietnam.  It doesn't make this incorrect necessarily, but does seem a bit contradictory.

Sometimes the words "most" or "majority" is used to describe things that are less than 50%.   Line 90 is example

Please include confidence intervals for all of your ORs. 

Most of the time 2 decimal points is acceptable for reporting ORs.

In the text, please indicate if these ORs were multivariate or univariate regression.  (Beginning line 101)

As mentioned above, please provide actual knowledge scores, not just "adequate/inadequate". 

Thank you so much for your comments. Please find our responses as followed:

- Regarding the geographic population, we did choose the sampling population based on the proportions of the total population of each province/area. However, the response rate of the people in the South of Vietnam was much higher than the rates in the Central and the North of Vietnam. Thus, the uneven results were obtained.

- About the “most” issue, we apologize in case we misunderstand your point. However, there is only one place in the manuscript that has the word “most” describing things less than 50%, as “Moreover, most of them were skilled workers (28.9%) and have medical insurance (95.3%).” Nevertheless, due to the fact that this occupation variable had 6 options (Student, Non-skilled worker, Skilled worker, Professional and managerial, Housewife, and Unemployed), the highest proportion was skilled workers, with 28.9% of the total participants. Thus, we humbly think that it is Ok to use the word “most” in this context.

- The confidence intervals values were added after all ORs data.

- Regarding the OR decimal points, we think that 3 decimal points would be more accurate. Thus, we are sincerely sorry for not following your suggestion on this point.

- The multivariate-analyses phrase have been added in the manuscript and Table 3, where appropriate.

-  Regarding the knowledge score, we apologize in case we misunderstand your point, in our humble opinion, we think that it is better to dichotomize the results. It was due to the fact that the score of each respondents were different, and the amount of respondents was high, thus, the continuous score presentation would be redundant and confusing, with lots of numbers. Conclusively, we thought that the grouping of this variable should be appropriate. 

Tables/Figures:

I would move the survey tool to the end or include as a supplemental, it interrupts the flow of the paper currently.

B13 in the survey questions, controlled is misspelled

Not your fault per se, but Tables 2 and 3 would be much better if they were on a single page each for layout for ease of reading.

Please indicate in Table 3 if this is univariate or multivariate; also include confidence intervals

Thanks for your comments. We would like to response as follows:

- Regarding the formatting issue, the whole article will be re-format again, in case it is accepted, by the Journal editorial team. Thus, we think the survey tool, Table 2 and 3, together with other items that need re-arrangement, should be adjusted later.

- The statement B13 was adjusted. Thank you.

- The data presented in Table 3 are multivariate analysis. We have indicated this in the Table name, we also included the confidence intervals in the Table.

Discussion:

Beginning at line 150, you talk a lot about over-the-counter pharmacy sales. This is fine, but it seems to be out of place in this initial paragraph. You quote a % of OTC purchases that was from a different study (156-158), then later (170) you quote your current data.  I would place these types of statements together to improve flow through this section.  There are some other opportunities to improve flow through these 2 paragraphs.

Is antibiotic quality a concern in Vietnam?  Please state either way.

Thanks a lot! These sentences and data have been re-arranged critically, where appropriate.

Line 165-167: You state the majority of patients seek advice before getting an antibiotic.  However that 59% means 41% do not, that's concerning.  I would reverse the way this is stated and focus on the 41%, not the 59%.  Again will help flow of paragraphs here.

For this issue, to be honest, we could not confirm due to the lack of literature and research regarding this in Vietnam. Moreover, we humbly think that this manuscript focuses on the antibiotics use and ABR, and although antibiotic quality is a crucial issue, it does not completely fit in the context of our research. Thus, we sincerely apologize for not following your comment. 

Thank you very much. We have revised these sentences following your suggestions.

Are antibiotics freely available in a pharmacy or have to distributed by a pharmacist (or pharmacy employee)?  That impacts access too

In Vietnam, antibiotics are freely available in a pharmacy, and patients can easily buy them without the needs of doctor prescriptions. This information has been added in the Introduction. Thanks for your question.

What is the role of pharmacies in Vietnam?  I would spend some more time discussing pharmacists/pharmacy role/responsibilities - now and perhaps future.  The questions infer there is limited impact of a pharmacist/pharmacy employee.  This would help the reader understand the landscape a bit better.

Also, opportunity to compare self-medication to other studies.  Haiti, PMID: 29138650; Tanzania, PMID: 32599699 (investigators actually went to community pharmacies to evaluate OTC).  I think leveraging pharmacies is an important discussion point to include that could be a next step for your research. 

Thanks for your suggestion. More information regarding this issue have been added in the Discussion.

Some of the characteristics that are linked to knowledge likely don't matter that much.  For example, while females scored statistically higher, adequate knowledge was in 26% vs 23% in females versus males.  I would argue that it doesn't really matter what's statistically associated when both are far lacking in knowledge based on your assessment.  Unless there were major differences in median scores. Please update the language to reflect this. 

Thank you for your comment. Regarding the associations between the socio-demographic characteristics and the respondents’ knowledge/practices, we only discussed the statistically significant figures. To this end, although some characteristics might seem to not really matter, we tried to discuss their relevance as much as possible. We do not want to rule out any significant values. We hope you understand our opinions and responses.

The survey tool itself has some major limitations that need to be mentioned.  - Many of the knowledge questions are very leading using the word "never" consistently.  This creates potential bias in answers.

- The survey did not appear to have a neutral response or an opportunity to not answer or defer on the question. That's a minor limitation.

- My experience has been that in these types of surveys people do not want to disappoint the surveyor and thus the accuracy of such questions like sharing antibiotics or not adhering to prescription instructions can be influenced.  However, it doesn't appear to be in your study, but still a potential limitation.

- Questions like D1 are a bit concerning of the overall message.  While I know the authors discourage self-prescribing, and thus self-diagnosis, this question actual calls for the respondent to self-diagnosis.  Unfortunately, it reflects a bit of the problem.  I think that question itself is a limitation.

Thank you. The survey tool has been critically developed, validated, and approved by the University of Malaya Research Ethics Committee. Moreover, the questionnaire was initially developed in English, and then translated into Vietnamese. The translated version was reviewed by secondary translators and independent language experts to ensure the accuracy of translation. Then, back translated version was double-checked by researchers for further modifications and improvements. The questionnaire was also validated by a panel of experts including physicians, medical microbiologists, academicians, and infection disease specialist. A pilot test with 20 eligible participants was additionally conducted to assess the interviewee’s understanding of sentence’s structure and wording. Thus, with this process, although the tool might not be perfect, we humbly think that it is Ok to use in the research. Nevertheless, we have added more limitations of the tool in the Limitations section.

- Does using households with a landline impact distribution at all?

Thanks for your question. Since almost all households in Vietnam have the landline telephone, we humbly think that this approach did not cause bias in the sample distribution.

Round 2

Reviewer 1 Report

The Second version of the manuscript has improved.

Is now better to understand and the results better discussed.

I sugest to change Table 1 to materials and methods section, after line 247.

Author Response

To: Antibiotics

Dear Editor and Editorial staffs,

We would like to send the revision letter regarding to the manuscript entitled “Factors influencing antibiotics overuse and misuse: A cross-sectional survey in Vietnam”, submitted to the Antibiotics, by Khanh Nguyen Di et al. We sincerely appreciate the hard work of the journal editors and the reviewers, which definitely makes our manuscript become better. Please find the responses to reviewers’ comments in the next pages of this letter. All changes in the manuscript were made under “track change” mode.

Thank you very much for your consideration. In case more information is required, please feel free to contact us.

Best regards,

Khanh Nguyen Di,

Reviewer 2 Report

The authors have addressed most comments presented by reviewers, thank you for that.  There are still limitations in the survey tool itself with lack of a neutral response and some of the questions themselves are leading.  I still feel this should be mentioned in the limitations section.  

Author Response

(The authors gave the same response as above.)
